# Longitudinal ^1^H NMR-Based Metabolomics in Saliva Unveils Signatures of Transition from Acute to Post-Acute Phase of SARS-CoV-2 Infection

**DOI:** 10.3390/v16111769

**Published:** 2024-11-13

**Authors:** Luiza Tomé Mendes, Marcos C. Gama-Almeida, Desirée Lopes Reis, Ana Carolina Pires e Silva, Rômulo Leão Silva Neris, Rafael Mello Galliez, Terezinha Marta Pereira Pinto Castiñeiras, Christian Ludwig, Ana Paula Valente, Gilson Costa dos Santos Junior, Tatiana El-Bacha, Iranaia Assunção-Miranda

**Affiliations:** 1LaRIV-Laboratory of Cellular Response to Viral Infections, Instituto de Microbiologia Paulo de Góes, Departamento de Virologia, Universidade Federal do Rio de Janeiro (UFRJ), Rio de Janeiro 21941-902, Brazil; luizatome12@gmail.com (L.T.M.); acpiresesilva@gmail.com (A.C.P.e.S.); romulo.lsneris@gmail.com (R.L.S.N.); 2LeBioME-Bioactives, Mitochondrial and Placental Metabolism Core, Institute of Nutrition Josué de Castro, Universidade Federal do Rio de Janeiro (UFRJ), Rio de Janeiro 21941-902, Brazil; marcosalmeidajj@gmail.com (M.C.G.-A.); desireellopes@gmail.com (D.L.R.); 3Núcleo de Enfrentamento e Estudos de Doenças Infecciosas Emergentes e Reemergentes (NEEDIER), Universidade Federal do Rio de Janeiro (UFRJ), Rio de Janeiro 21941-599, Brazil; galliez77@gmail.com (R.M.G.); tmartapc@gmail.com (T.M.P.P.C.); 4Department of Metabolism and Systems Science, School of Medical Sciences, College of Medicine and Health, University of Birmingham, Birmingham B15 2TT, UK; c.ludwig@bham.ac.uk; 5National Center for Nuclear Magnetic Resonance—Jiri Jonas, Institute of Medical Biochemistry, Universidade Federal do Rio de Janeiro (UFRJ), Rio de Janeiro 21941-902, Brazil; valente.anap@gmail.com; 6LabMet-Laboratory of Metabolomics, Instituto de Biologia Roberto Alcantara Gomes (IBRAG), Department of Genetics, State University of Rio de Janeiro, Rio de Janeiro 20551-030, Brazil; gcostadossantosuerj@gmail.com

**Keywords:** metabolome, saliva, SARS-CoV-2 infection, long COVID, metabolic fluctuation

## Abstract

COVID-19 can range from a mild to severe acute respiratory syndrome and also could result in multisystemic damage. Additionally, many people develop post-acute symptoms associated with immune and metabolic disturbances in response to viral infection, requiring longitudinal and multisystem studies to understand the complexity of COVID-19 pathophysiology. Here, we conducted a ^1^H Nuclear Magnetic Resonance metabolomics in saliva of symptomatic subjects presenting mild and moderate respiratory symptoms to investigate prospective changes in the metabolism induced after acute-phase SARS-CoV-2 infection. Saliva from 119 donors presenting non-COVID and COVID-19 respiratory symptoms were evaluated in the acute phase (T1) and the post-acute phase (T2). We found two clusters of metabolite fluctuation in the COVID-19 group. Cluster 1, metabolites such as glucose, (CH_3_)_3_ choline-related metabolites, 2-hydroxybutyrate, BCAA, and taurine increased in T2 relative to T1, and in cluster 2, acetate, creatine/creatinine, phenylalanine, histidine, and lysine decreased in T2 relative to T1. Metabolic fluctuations in the COVID-19 group were associated with overweight/obesity, vaccination status, higher viral load, and viral clearance of the respiratory tract. Our data unveil metabolic signatures associated with the transition to the post-acute phase of SARS-CoV-2 infection that may reflect tissue damage, inflammatory process, and activation of tissue repair cascade. Thus, they contribute to describing alterations in host metabolism that may be associated with prolonged symptoms of COVID-19.

## 1. Introduction

Coronavirus disease 2019 (COVID-19), caused by severe acute respiratory syndrome coronavirus 2 (SARS-CoV-2) infection, continues to be a challenge to the healthcare system in several countries [1]. Since COVID-19 emergence, in addition to vaccination, several efforts have been made to improve the diagnosis and prognosis of severe and critical cases and to control the spread of the infection [2]. However, due to high viral transmission rates, new SARS-CoV-2 genetic variants of concern are still emerging, resulting in the maintenance of viral spread, affecting vaccine performance and altering the pattern of COVID-19 clinical manifestation [2,3].

It is now well established that the pathogenesis of SARS-CoV-2 is multisystemic [4,5]. Although the majority of cases are mild to moderate and the mortality rate ranges from 2 to 5% of symptomatic cases, it is estimated that at least 65 million people have developed post-acute COVID-19 symptoms. This number is probably higher due to undocumented or misdiagnosed instances [1,6]. The persistence of symptoms for more than 3 to 4 weeks after acute SARS-CoV-2 infection has been classified as “long COVID” or “post-acute COVID-19 syndrome” [7,8]. The clinical manifestations of long COVID include muscular, cardiovascular, neuronal, and endocrine disorders, reflecting the impact of SARS-CoV-2 infection on multiple organs at the acute phase [7,8]. Importantly, these symptoms were not restricted to subjects that developed severe COVID-19, and they are also observed after mild and moderate cases, and even in asymptomatic infections [6,9]. These data indicate that virtually all cases of SARS-CoV-2 infection represent a risk of developing some long COVID sequelae. However, the mechanisms associated with the clinical evolution to post-acute symptoms are poorly understood.

The pathophysiology of long COVID may be related to immune and metabolic responses induced by acute infection [10,11]. Metabolomics studies have associated the disturbances triggered by acute SARS-CoV-2 infection with alterations in plasma metabolites, such as lipids, glucose, one carbon, and mainly amino acids. These systemic alterations reflect the extension of inflammation, lesion, and dysfunction of target tissues, ranging from moderate to fatal cases of COVID-19, and they could be used as biomarkers for the prognosis of disease outcomes in the acute phase [12,13,14,15]. In addition, the development of insulin resistance, diabetes, and other metabolic disturbances observed in the post-acute phase indicates that metabolic dysfunction may persist even after viral clearance [12,16,17,18].

The characterization of the saliva metabolome can also help describe the metabolic pathways involved in the clinical evolution of COVID-19 [19,20]. The epithelial cells of the oral mucosa and salivary gland express the cellular receptor angiotensin-converting enzyme 2 (ACE2), and they are potential sites for the initial replication of SARS-CoV-2 [20]. Thus, saliva metabolites may reflect direct changes in the target sites of infection, derived from activated immune cells in the mucosa and oral microbiota responses [20]. Furthermore, changes in saliva metabolism also follow metabolite fluctuation in blood, reflecting systemic alterations in pathophysiological conditions [20,21].

The literature is still scarce when it comes to longitudinal evaluation of the metabolic responses related to the post-acute phase of COVID-19, mainly in subjects with mild and moderate disease. In the present study, we conducted a ^1^H Nuclear Magnetic Resonance (NMR) metabolomics study in symptomatic subjects presenting mild and moderate respiratory symptoms to investigate longitudinal changes in saliva metabolome between the acute and the post-acute phases of SARS-CoV-2 infection. Moreover, we investigated how these metabolic alterations observed in COVID-19 subjects were associated with the degree of replication, vaccination status, and the presence of comorbidities. Our approach contributes to the description of post-acute alterations in host metabolism that may be associated with long COVID.

## 2. Materials and Methods

### 2.1. Study Design and Participants

Recruitment took place between April and July 2021 at the Center for COVID-19 diagnosis, presently Núcleo de Enfrentamento e Estudos de Doenças Infecciosas Emergentes e Reemergentes (NEEDIER) at the Federal University of Rio de Janeiro, Rio de Janeiro, Brazil.

At the time of admission, 119 symptomatic adults (≥18 years of age) were included in this study 2–5 days after the beginning of symptoms after the screening test for viral antigen. SARS-CoV-2 diagnosis was confirmed by nasopharyngeal swab Real-Time PCR (RT-qPCR) [22]. Viral RNA was extracted using the Maxwell 16 viral total nucleic purification kit system (Promega, Madison, WI, USA) according to the manufacturer’s instructions. Viral RNA was detected using the SARS-CoV-2 (2019-nCoV) CDC qPCR probe assay (Integrated DNA Technologies, Coralville, IA, USA), targeting the SARS-CoV-2 N1 and N2 genes and the human RNase P gene. All reactions were paired and performed in a 7500 Thermal Cycler (Applied Biosystems, Foster City, CA, USA). A SARS-CoV-2 RT-qPCR result was considered positive if both targets (N1 and N2) were amplified with a cycle threshold (Ct) value ≤ 37.

Subjects were grouped into SARS-CoV-2 positive (n = 63; COVID-19 group) and SARS-CoV-2 negative, presenting non-COVID respiratory symptoms (n = 56; non-COVID group). Subjects returned 45 days after the first visit, corresponding to n = 32 in the COVID group and n = 20 in the non-COVID group (Figure 1A). The first visit corresponded to the acute phase of COVID-19 (T1) and the second visit corresponded to the post-acute phase of COVID-19 (T2). In both T1 and T2, saliva samples were collected in the morning (from 8 a.m. to 11 p.m., local time) through auto sampling, and clinical and demographic data and previous diagnosis of COVID-19 were obtained via structured forms.

### 2.2. Ethics Approval

The present study was approved by the local ethics review committee from Clementino Fraga Filho University Hospital (CAAE: 30161620.0.0000.5257) and by the national ethical review board (CAAE: 30127020.0.0000.0068). Informed consent was obtained from all subjects involved in this study.

### 2.3. Saliva Sampling and Processing

For saliva sampling, participants were asked to briefly wash their mouth with water for prior hygiene of the oral cavity, and sampling was performed by spontaneous salivation in a sterile universal collector containing 1.25% Triton X-100 (Sigma-Aldrich, St. Louis, MO, USA) for viral inactivation. Only participants who did not consume food up to one hour before the collection were included in this study. Samples (approximately 1 mL of each participant) were immediately transferred to Eppendorf tubes and centrifuged at 4 °C at 14,000× *g* for 45 min to remove cellular debris. All samples were stored at −80 °C until the analysis.

### 2.4. NMR-Based Metabolomics

#### 2.4.1. Sample Preparation and NMR Acquisition

Saliva samples were thawed and diluted 1.2-fold in a sodium phosphate buffer solution and deuterium oxide to a final concentration of 50 mM phosphate buffer, 16% deuterium oxide, and 0.2 mM DSS, pH 7.4. A total of 600 μL of diluted samples were transferred to the 5 mm NMR tube.

NMR spectra were acquired on a Bruker Advance III spectrometer at 500 MHz with a temperature of 298 K coupled to an automatic sample changer and cooled to 280 K. One-dimensional ^1^H spectra were acquired with suppression of the water signals using the excitation sculpting pulse sequence [23]. Broad resonances from the protein background were suppressed using the Carr–Purcell–Meiboom–Gill pulse sequence (CPMG) [24] as a transverse relaxation filter with 32 loop counters and a delay of 0.001s. Therefore, the effective T2 delay of the transverse relaxation filter was 68.68 ms; 32,768 complex data points were acquired per transient, for a total of 1024 transients. The spectral width was set to 19.99 ppm, resulting in an acquisition time of 3.27 s per free induction decay (FID). The relaxation delay was set to 1.74 s.

#### 2.4.2. NMR Spectra Pre-Processing and Assignment 

Spectra were pre-processed using MetaboLab software v. 2022.0726.1733 [25]. Prior to the Fourier transform, the FIDs were apodized using an exponential window function with 0.3 Hz line broadening and were then zero filled to 131,072 real data points. After the Fourier transform, each spectrum was manually phase corrected, followed by a spline–baseline correction. Spectra were referenced to the DSS signal. Signal-free regions (0.29 to −5.2 and 14.79 to 9.73), water signals (5.16 to 4.59 ppm), ethanol signals (1.08 to 1.29 ppm), and triton signals added for viral inactivation (8.61 to 8.26, 7.25 to 7.09, 6.86 to 6.60, 3.72 to 3.54, 2.94 to 2.80, 2.21 to 2.13, 1.71 to 1.52, 1.08 to 1.01, 0.8 to 0.3 ppm) were excluded. Additionally, noise filtering was performed, and anything below 5 times the noise threshold—measured between 9.5 and 10 ppm region—was discarded. Spectra were aligned using the Icoshift algorithm [26], which is integrated into Metabolab software.

Spectral data were then binned with a 0.005 ppm interval (32 data points), as this ppm interval while correcting for small peak shifts and promoting spectral smoothing keeps all spectral information. The resulting table presented 1132 variables/bins. For the comparison of each individual metabolite between groups, metabolites’ intensity was used after performing spectral binning. The spectral bins that were chosen for comparison were the ones with the highest intensity, accounting for the whole peak, and also had no overlapping signals.

All spectra were normalized using probabilistic quotient normalization, and Pareto scaling was applied prior to multivariate statistics. The TOCSY ^1^H-^1^H was uploaded on the COLMAR [27] for the assignments. The peak report of all assigned compounds can be seen at https://spin.ccic.osu.edu/index.php/colmarm, session ID 1399-nBZFhqa0aQ, (accessed on 31 March 2022). The Human Metabolome DataBase (HMDB) [28] and Chenomx NMR Suite 8.2^®^ program (Chenomx Inc., Edmonton, AB, Canada) were also used for the assignment of metabolites. Appendix A presents the ^1^H NMR assignment information for the metabolites.

### 2.5. Statistical Analysis 

Data distribution was analyzed using the Shapiro–Wilk test, and continuous variables with non-parametric distributions were represented as medians and compared using the Mann–Whitney test. Categorical variables were compared using the chi-square test with absolute (n) and relative (%) frequencies.

The processed NMR spectra data were analyzed using the MetaboAnalyst platform [29], applying multivariate statistical principal component analysis (PCA) and a heatmap. The heatmap was generated with normalized data using the Euclidean distance measure, the clustering method Ward, and the variance IQR method. All metabolites found in this study were subjected to Metabolite Set Enrichment Pathway Analysis (MSEA), available on the Metaboanalyst 5.0 [29] platform, and were compared with the Kyoto Encyclopedia of Genes and Genomes (KEGG) database. The affected pathways were deemed significant when *p* < 0.05.

For univariate statistics, Mann–Whitney and Kruskal–Wallis tests were used to compare non-transformed NMR data. Graph Pad Prism^®^ software version 8.0.1. was used for all analyses. *p* values < 0.05 were considered to reject the null hypothesis.

## 3. Results

### 3.1. Subjects’ Demographic Characteristics and Clinical Parameters

Subjects in the COVID-19 group were significantly older than the non-COVID respiratory symptoms (non-COVID) group, while sex distribution was similar (Table 1). Most symptoms were more prevalent in the COVID-19 group, with highlights to fever, cough, adynamia, anosmia, and ageusia (Table 1). Regarding the duration of symptoms in the COVID-19 group, 44.4% had symptoms for up to 15 days and 19% for 30 days or more. The prevalence of comorbidities was similar between groups, and almost half of the subjects in both the COVID-19 and non-COVID groups presented as being overweight or obese. The majority of subjects in both groups were unvaccinated (~71% in the non-COVID and ~63% in the COVID-19 groups).

### 3.2. Saliva NMR-Based Metabolomics Shows Specific Metabolite Fluctuation from the Acute to the Post-Acute Phase of COVID-19

Representative spectra of T1 and T2 samples indicated differences in metabolites in the aliphatic, amide, and aromatic regions when comparing the acute and post-acute phases, mainly in the COVID-19 group (Figure 1B–E). Metabolites with differential signals, such as 2-hydroxybutyrate, (CH_3_)_3_ choline-related metabolites, sugar regions (mainly glucose), fumarate, phenylalanine, and histidine, are indicated in the figures. The heatmap plot confirmed the differences between T1 and T2 in each group and revealed two clusters of changes in metabolites. Cluster 1 comprised metabolites that increased in T2, and cluster 2 comprised metabolites that decreased in T2 compared to T1 levels in the COVID-19 group. In the non-COVID group, there is no variation in the levels of metabolites in cluster 1, and the pattern of metabolite changes in cluster 2 was opposite to what was observed in the COVID-19 group (Figure 1F, Appendix A). In cluster 1, glucose, (CH_3_)_3_ choline-related metabolites, 2-hydroxybutyrate, and BCAA stood out, and in cluster 2, acetate and lysine contributed to this distinct pattern in T2 compared to T1. Despite the PCA score plot did not indicate a discriminant pattern between T1 and T2 in both groups (Appendix A), the PCA loading factors plot confirmed the heatmap results, where changes in glucose, 2-hydroxybutyrate, BCAA, and lactate contributed to the profile between the acute and post-acute phases of COVID-19 (Appendix A). 

Univariate statistics were used to gain meaningful insights into the changes associated with the transition from acute to post-acute phases of COVID-19. We selected primary discriminating metabolites based on the PCA loading factors (Appendix A) and heatmap results (Figure 1F, Appendix A), and their levels were plotted as T2 relative to T1 in the COVID-19 and non-COVID groups (Figure 2). In the COVID-19 group, a significant increase in T2 compared to T1 in the levels of glucose, fumarate (a tricarboxylic cycle acid metabolite), (CH_3_)_3_ choline-related metabolites (choline, glycerophosphocholine), 2-hydroxybutyrate, and the amino acids BCAA and taurine were found (Figure 2B). In contrast, creatine/creatinine and lysine/putrescine (considered together due to signal overlap), histidine, phenylalanine, and acetate decreased in T2 relative to T1 (Figure 2B). Enrichment pathway analysis indicated that the most affected metabolic pathways in the transition from the acute to post-acute phase in the COVID-19 group were amino acids, lipids, the TCA cycle, and glucose metabolism (considering *p* < 0.05 values) (Appendix A). Importantly, in the non-COVID group, only lactate was reduced in T2 compared to T1 (Figure 2A), which was not altered in the COVID-19 group (Figure 2B). Taken together, our data indicate that the clinical evolution to the post-acute phase in COVID-19 subjects involves metabolic changes differing from subjects that present non-COVID respiratory symptoms. 

### 3.3. Longitudinal Changes in Metabolites in the COVID-19 Group Were Associated with BMI and Vaccination Status, but Not with Biological Sex

Longitudinal changes in metabolites in the COVID-19 group were also evaluated as a function of the body mass index (BMI), as excessive body weight is a risk factor for COVID-19 severity. COVID-19 subjects were classified as eutrophic (BMI < 25 kg/m^2^) or overweight/obese (BMI ≥ 25 kg/m^2^) (Figure 3A,B). In eutrophic subjects, no significant differences were found in metabolites when comparing T2 relative to T1 (Figure 3A); whereas, in the overweight/obese group, an increase in fumarate, (CH_3_)_3_ choline-related metabolites, 2-hydroxybutyrate, taurine, and BCAA and a decrease in acetate, histidine, and creatine/creatinine were observed (Figure 3B).

Vaccination status was also considered to evaluate the longitudinal changes in metabolites in the COVID-19 group, and subjects were grouped into unvaccinated and vaccinated (received at least one dose of the vaccine). In the vaccinated group, only BCAA presented a longitudinal increase (Figure 3C). Interestingly, in the unvaccinated group, there was a decrease in amino acids and acetate, and an increase in glucose, fumarate, (CH_3_)_3_ choline-related metabolites, and taurine in T2 (Figure 3D). Moreover, when stratifying COVID-19 subjects according to biological sex, few longitudinal variations in metabolites were observed (Appendix A). 

### 3.4. Longitudinal Changes in Metabolites from the Acute to Post-Acute Phase Were Associated with SARS-CoV-2 Replication

We then investigated whether the metabolic alterations observed in T2 in the COVID-19 group could be associated with viral load detected in the respiratory tract during the acute phase. For this purpose, qPCR Ct (cycle threshold) values were used to classify subjects that presented higher viral load (high; Ct < 18.9 cycles) or lower viral load (low; Ct > 18.9 cycles) (Figure 4A,B). In the subjects classified as low (Figure 4A), only feeble longitudinal changes were detected in T2 compared to T1, as opposed to what was observed in the high group, with several metabolites changed in T2 (Figure 4B). It is important to mention that increased levels of BCAA in T2 were only observed in the low group (Figure 4A). Lastly, subjects in the COVID-19 group were stratified according to the persistence of detection of the SARS-CoV-2 genome in the respiratory tract. Based on the qPCR test: they were positive for up to two weeks or positive for more than three weeks (Figure 4C,D). Metabolite fluctuations were observed in subjects that remained positive for SARS-CoV-2 for up to 2 weeks (Figure 4C).

## 4. Discussion

This study intended to identify metabolic markers in the saliva associated with the clinical evolution of COVID-19 and to explore the contributing factors involved in the metabolic dysregulation caused by SARS-CoV-2 infection. Saliva metabolome reflects systemic and in situ metabolism associated with the disease state, and the contribution of oral microbiota must be considered. We identified two clusters of metabolites in the transition from the acute to post-acute phase that distinguished COVID-19 subjects from those that presented non-COVID respiratory symptoms. These metabolites are enrolled in the metabolism of amino acids, lipids, the TCA cycle, and glucose. Importantly, these longitudinal changes evidenced in our study contribute to the description of a metabolic signature involved in the tissue repair cascade but also unveil some persistent metabolic imbalances associated with late symptoms of COVID-19.

A set of changes in metabolites observed in our study may be associated with poor recovery and can be regarded as long COVID responses, including the decrease in histidine and acetate, and the increase in 2-hydroxybutyrate and glucose levels. Histidine is a substrate for the synthesis of histamine by immune cells, regulating physiological and unbalanced immune responses in target cells [30,31]. The significant reduction in histidine levels may be a metabolic response of persistent inflammation due to an increased histamine synthesis by macrophages, dendritic cells, and T lymphocytes [30,32]. In agreement, it has been reported that excessive histamine production may decrease plasma histidine levels in COVID-19 subjects, contributing to rashes and inflammation seen during the disease [30]. Indeed, the local release of histamine may regulate cytokine production by different immune cells, associated with inflammation observed in COVID-19 subjects [30,31,32]. Likewise, the reduction in acetate in the saliva was associated with hospitalized subjects due to complications in the post-acute phase of COVID-19 [33]. Acetate is produced by the microbiota, and it has been shown to improve B cell metabolism involved in the production of specific antibodies against SARS-CoV-2, playing a crucial role in the control of infection [34]. These results may reflect dysregulations in the gut and oral microbiota observed in the acute phase of SARS-CoV-2 infection [34,35,36], which probably persist in long COVID [37]. Interestingly, the reduction in histidine and acetate levels was observed mainly in overweight/obesity subjects, which have been considered a risk factor for the progression to the severe forms of COVID-19 [38] and also in unvaccinated subjects, reinforcing the protective effect of vaccination in the acute and long-lasting inflammatory process in COVID-19 [39,40].

Increased levels of 2-hydroxybutyrate and glucose in the circulation were also correlated with acute inflammatory responses in COVID-19 subjects [41,42,43]. A study by Barreto et al. demonstrated that COVID-19-related hyperglycemia is a result of a PEPCK-dependent gluconeogenic effect in infected hepatocytes [44]. Thus, the persistent hyperglycemic environment evidenced by our study may be indicative of compromised tissue functions, including the liver, and decreased insulin sensitivity [11,12,31]. Similar to what we observed, the development of hyperglycemia occurs irrespective of sex, BMI, and other comorbidities, such as diabetes mellitus [44]. Increased glucose levels and other markers of persistent inflammation were observed in subjects with higher viral load and were positive for up to 2 weeks, indicating that hyperglycemia is associated with SARS-CoV-2 replication [45]. In agreement, previous studies have associated SARS-CoV-2 replication with the extension of the induced inflammatory response and disease outcome [45]. In addition, hyperglycemia has also been correlated with higher viral load in COVID-19 subjects and directly impacts viral replication, ACE2 expression, and cytokine production by peripheral blood monocytes infected with SARS-CoV-2 in vitro [41]. Therefore, it can be suggested that the magnitude of the metabolic changes in the post-acute phase was influenced by the degree of the inflammatory response and viral replication in the acute phase of infection [8,10,11]. Interestingly, Heald et al. showed that vaccination against COVID-19 had a positive effect on regulating blood glucose levels in subjects with diabetes mellitus [46].

On the other hand, we observed some longitudinal metabolic changes in the post-acute phase of COVID-19 that could be involved in the resolution of inflammation and also the tissue repair cascade. Specifically, the increase in fumarate, BCAA, taurine, and choline and the decrease in creatine/creatinine levels could be potential markers of protection. Fumarate and its derivatives, such as dimethyl fumarate, are potent modulators of anti-antioxidant responses that inhibit SARS-CoV-2 replication and induce inflammation [47,48,49,50]. In addition, low levels of BCAA have been associated with systemic manifestations and fatal cases of COVID-19 [12,51]. Indeed, taurine and BCAA supplementation have been proposed as a strategy to attenuate skeletal muscle wasting and mitigate tissue damage in COVID-19 subjects [30,52,53]. In our study, BCAA levels were only increased in subjects with low viral load, reinforcing that BCAA are potential markers and protector factors for the clinical evolution of COVID-19. Higher levels of creatine/creatinine were also associated with pro-inflammatory cytokines, such as IL-6, in subjects with severe COVID-19 [54,55] and also with COVID-19 adrenal dysfunction and mortality [12,54]. Therefore, the reduction in creatine/creatinine levels suggests some degree of tissue recovery in the post-acute phase. It is also worth highlighting that creatine/creatinine dysregulation is recognized as a facilitator of viral replication [12,56], which is consistent with the higher levels of this metabolite during an acute phase. The increase in choline levels in the post-acute phase of COVID-19 subjects can also be considered a protective response. A reduction in choline and its derivatives in the plasma of subjects with severe COVID-19 and in individuals with mild COVID-19 infected with SARS-CoV-2 omicron [12,57,58] have been associated with reduced synthesis of phospholipids, liver fibrosis, alterations in the gut microbiota, and thrombotic events [12].

The novelty/strength of our work was to show that there are important changes in these metabolites in the post-acute compared to the acute phase of SARS-CoV-2 infection. It is important to draw attention to the fact that, in general, the metabolic signature linked to tissue repair or inflammation was present in the group with high viral load but not in subjects with persistent detection of SARS-CoV-2. This indicates that the magnitude of the inflammatory response and persistent metabolic changes induced by viral replication are linked to viral clearance and the activation of tissue repair pathways. Thus, it is possible that long COVID symptoms are determined by persistent inflammation but are also a result of activation of the repair cascade. Our work showed that saliva is a potential biofluid to address these aspects. It is not clear whether our findings correlate with the severity of the disease, as only a small proportion of subjects in our study had prolonged symptoms, and additional studies are necessary. In addition, longitudinal studies evaluating plasma metabolites would contribute to differentiating metabolic markers of the oral/respiratory replication site from those that are observed systemically.

## Figures and Tables

**Figure 1 viruses-16-01769-f001:**
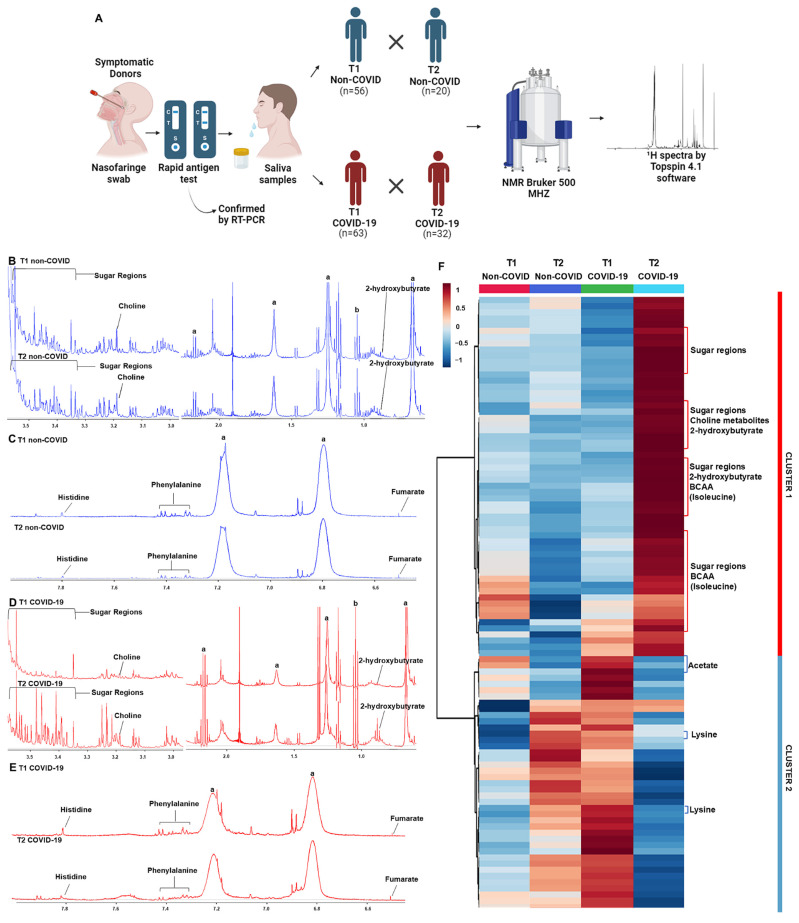
^1^H NMR-based metabolomics shows altered saliva metabolite profiling in the post-acute phase of COVID-19. (**A**) Experimental study design showing that saliva samples were collected from subjects with non-COVID respiratory symptoms (non-COVID) and symptomatic subjects infected with SARS-CoV-2 (COVID-19) in two moments: acute phase (T1) and post-acute phase (T2). Representative ^1^H NMR spectra of T1 and T2 phases from non-COVID and COVID-19 groups: (**B**,**C**) aliphatic, amidic, and aromatic regions from the non-COVID group; (**D**,**E**) aliphatic, amidic, and aromatic regions from the COVID-19 group; symbols correspond to peaks of a, triton, and b, ethanol. (**F**) The heatmap shows two distinct clusters of metabolic profiles in the samples of non-COVID and COVID-19 groups between T1 and T2.

**Figure 2 viruses-16-01769-f002:**
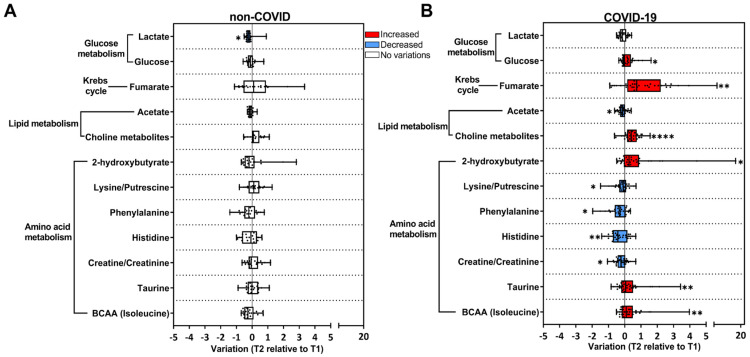
Univariate analysis of the metabolic alterations in saliva reveals a signature associated with COVID-19. Longitudinal changes in metabolites are presented as the ratio T2 to T1, calculated from metabolite intensity according to ^1^H NMR metabolomics. Red bars: higher metabolite content in T2; blue bars: lower metabolite content in T2; white bars: no differences between T1 and T2 phases. (**A**) Non-COVID subjects (T1 n = 56; T2 n = 20); (**B**) COVID-19 subjects (T1 n = 63; T2 n = 32). * *p* < 0.05, ** *p* < 0.01, **** *p* < 0.0001, according to T1 and T2 comparison, unpaired Mann–Whitney test.

**Figure 3 viruses-16-01769-f003:**
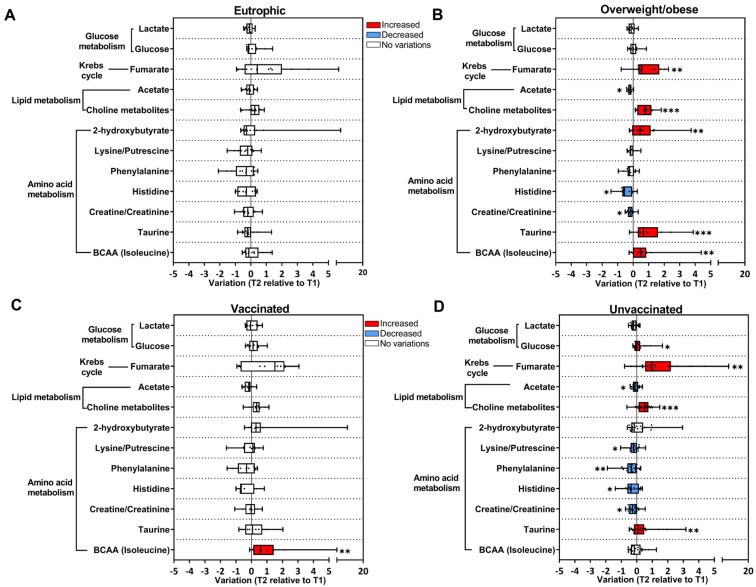
BMI and vaccination status contribute to longitudinal changes in metabolites in the COVID-19 group. Longitudinal changes in metabolites are presented as the ratio T2 to T1 calculated from metabolite intensity according to ^1^H NMR metabolomics. Red bars: higher metabolite content in T2; blue bars: lower metabolite content in T2; white bars: no differences between T1 and T2 phases. (**A**) Eutrophic subjects BMI < 25 kg/m^2^ (T1 n = 21; T2 n = 13); (**B**) overweight/obese subjects BMI > 25 kg/m^2^ (T1 n = 26; T2 n = 12); (**C**) vaccinated subjects (T1 n = 19; T2 n = 11); (**D**) unvaccinated subjects (T1 n = 40; T2 n = 21). * *p* < 0.05, ** *p* < 0.01, *** *p* < 0.001, according to T1 and T2 comparison, unpaired Mann–Whitney test.

**Figure 4 viruses-16-01769-f004:**
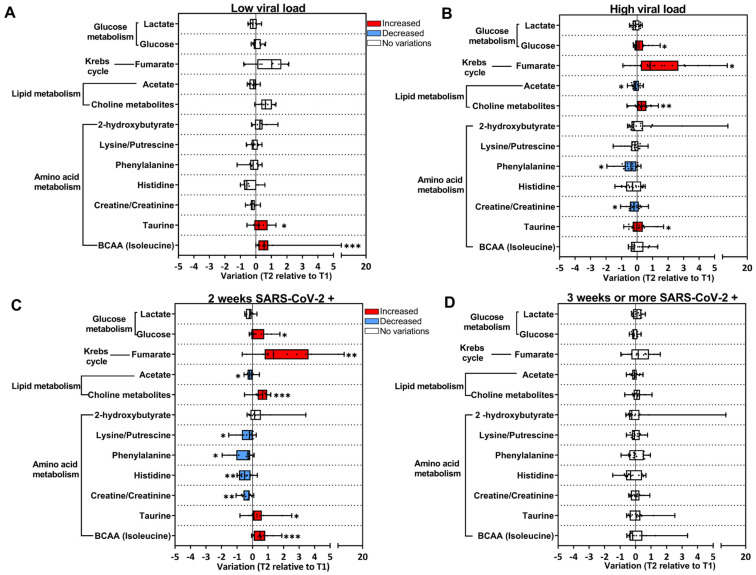
SARS-CoV-2 replication contributes to the alterations in the metabolic profile in the COVID-19 group. Longitudinal changes in metabolites are presented as the ratio T2 to T1, calculated from metabolite intensity according to ^1^H NMR metabolomics. Red bars: higher metabolite content in T2; blue bars: lower metabolite content in T2; white bars: no differences between T1 and T2 phases. (**A**) Subjects with lower viral load (T1 n = 25; T2 n = 11); (**B**) subjects with higher viral load (T1 n = 38; T2 n = 21); (**C**) SARS-CoV-2-positive subjects for up to 2 weeks (T1 n = 30; T2 n = 13); (**D**) SARS-CoV-2-positive subjects for up to 3 weeks or more (T1 n = 22; T2 n = 16). * *p* < 0.05, ** *p* < 0.01, *** *p* < 0.001, according to T1 and T2 comparison, unpaired Mann–Whitney test.

**Table 1 viruses-16-01769-t001:** Demographic characteristics of subjects in the non-COVID and COVID-19 groups.

	Non-COVID (n = 56) ^1^	COVID-19 (n = 63) ^1^	*p* Value ^2^
Age, years (median, min-max)	32 (17–62)	42 (20–70)	**0.0035**
Sex, female; n (%)	35 (62.5%)	35 (55.5%)	0.4423
**Symptoms: n (%)**			
Fever	19 (33.9%)	44 (69.8%)	**0.0001**
Headache	39 (69.6%)	44 (69.8%)	0.9812
Cough	30 (53.5%)	47 (74.6%)	**0.0166**
Sore throat	32 (57.1%)	31 (49.2%)	**0.0491**
Nasal congestion	40 (71.4%)	51 (80.9%)	0.2215
Chills	23 (41%)	34 (53.9%)	0.1598
Nausea or vomiting	11 (19.6%)	24 (38.1%)	**0.0275**
Myalgia	22 (39.2%)	42 (66.6%)	**0.0028**
Adynamia	29 (51.7%)	53 (84.1%)	**0.0028**
Anosmia and ageusia	6 (10.7%)	24 (38%)	**0.0006**
**Comorbidities: n (%)**			
Declared at least one	31 (55.3%)	41 (65%)	0.2789
Overweight/obesity	24 (42.8%)	26 (41.2%)	0.8610
Chronic respiratory disease	6 (10.7%)	4 (6.3%)	0.3916
Hypertension	8 (14.2%)	11 (17.4%)	0.6370
Diabetes mellitus	4 (7.1%)	2 (3.1%)	0.3234
N/D ^3^	11 (19.6%)	11 (17.4%)	0.7595
**Vaccination: n (%)**			
Unvaccinated	40 (71.4%)	40 (63.5%)	0.3572
Vaccinated (1 or 2 doses)	15 (26.7%)	19 (30.2%)	0.6841
N/D ^3^	1 (1.8%)	4 (6.3%)	0.2155
**Duration of symptoms: n (%)**			
15 days	n.a	28 (44.4%)	n.a
More than 30 days	n.a	12 (19%)	n.a
N/D ^3^	n.a	23 (36.5%)	n.a
**Time of positivity for SARS-CoV-2 in the RT-PCR test (n = 52): n (%)**			
2 weeks	n.a	30 (57.7%)	n.a
3 weeks or more	n.a	22 (42.3%)	n.a

^1^ Continuous variables are represented as median and categorical variables are represented as n (frequency %). n.a.—not applicable; ^2^ *p* values between non-COVID and COVID-19 groups. Categorical variables were compared using the chi-square test and continuous variables using the Mann–Whitney U test for non-parametric distributions; ^3^ N/D—not determined. Significant *p* values are in bold.

## Data Availability

The raw NMR data presented in this study are available on request from the corresponding author. Individual results used to univariate analysis were presented in dots at figures.

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
