# Peer review of "Longitudinal 1H NMR-Based Metabolomics in Saliva Unveils Signatures of Transition from Acute to Post-Acute Phase of SARS-CoV-2 Infection"

_viruses, 2024, doi:10.3390/v16111769_

Round 1
Reviewer 1 Report
Comments and Suggestions for Authors
view file

Author Response
The study is based on a valid scientific rationale and aims to provide new knowledge into a still unclear field such as long -COVID-19 effects. However, it presents a series of criticisms, especially from the point of view of statistics and data evaluation.
We thank the Reviewer for his/her comments. We considered the questions raised, clarified some points about the analyzes and evaluated possible outliers. All suggestions were incorporated in the new version of the manuscript.
1-It would be advisable to show in the Supplementary material binning data and to motivate the parameters used, such as 0.005 ppm range.
Response. We thank the reviewer for addressing these aspects. We obtained a total of 81,498 data points for all spectra that, after processing, resulted in a table with a final number of 1132 variables/bins. We now included a binning table of assigned metabolites in the supplementary material of the revised version (supplementary table S1).
Also, spectral data were binned with a 0.005 ppm interval (32 data points) as this ppm interval is often used in NMR-based metabolomics and in the case of our data, apart from reducing the number of variables in the data matrix, it significantly corrected for small peak shifts and also promoted spectral smoothing. Importantly, the 0.005 ppm range parameter kept all spectral information.
Comments of this nature were included in the revised version of the manuscript (lines 169-74).
2-The pathways analysis is not significant for a p.value value < 0.05; it is essential to consider also hits and Holm adjustment p value, as well as the Impact value.
Response. We agree that Holm adjustment p value and impact value are also important to support pathway analysis. However, we consider to maintain the analysis in supplementary material and use p-values < 0.05 as indicative of pathways in which these significantly altered metabolites between the acute and post-acute phase of COVID would be involved, but including all statistical data. Thus, we specified the parameter used for the statements in the result section of the manuscript main text (line 262) and changed the graphic to an “Enrichment pathway” analysis as suggested in the topic 9 (supplementary figure S3 and Table S3).
3-lines 18 Sample preparation and NMR acquisition: Probably it means 500 MHz and not 500.13. The spectrometer is not consistent with the recommended 600MHz frequency for metabolomics analysis.
Response. We thank the reviewer for addressing this aspect. The spectrometer frequency reported was 500.13 MHz, which also could be reported as 500 MHz. We changed this information in the revised manuscript (line 147).
Regarding the field strength and resolution, we are aware that the “sweet spot” in NMR-based metabolomics is the 600 MHz NMR spectrometer (10.3390/metabo9070123). However, we have access to a 500 MHz instrument, which should still provide sufficient resolution and sensitivity for metabolomics analyses and was used in previous studies by our group (10.3390/metabo13070879; 10.1021/acs.jproteome.3c00014). As a way to improve spectral resolution and metabolites assignment, we also ran TOCSY experiments, as described in the methods (lines 176-79).
4- Figure 1B-E: Figure 1B-E is not clear. It would be advisable to show the overlapping spectral windows and calculate a spectrum difference between the groups, especially for the areas that are significantly different.
Response. We apologize for the poor spectral quality which prevented the visualization of metabolites’ peaks. In this new version of the manuscript, we replaced figures 1B-E in order to present a better spectral superposition and to clearly point out the regions where we identified differences in metabolites peaks height intensity, between T1 and T2 spectra. Since these spectra were chosen to visually show the peak differences of the indicated metabolites, we decided not to include in the figure the actual difference between areas. Considering the improved spectral quality of this new Figure 1, in our opinion, the inclusion of the spectrum difference after overlapping would not add to the discussion since the peak intensity data obtained were used to compare the groups using further multivariate and univariate analysis, and individual fluctuations of metabolites were shown in Figure 2.
It is important to mention that, in agreement with what was described in the study, differences can be observed mainly in the COVID group, which are depicted in the representative spectra. Importantly, the same metabolites were described with highest loading factors from PCA (Supplementary Table S2), were found in the heatmap (Figure 1 F) and presented a significant fluctuation in T1 from T2 in univariate analysis (Figure 2).
5-Figure 1F: It would be desirable to show the clustering in the heatmap as well as the whole dataset, ( not only the averages) to understand if the concentrations of the discriminating metabolites are representative of the population under study.
Response. We thank the reviewer for his/her suggestion. We included in the supplementary material a heatmap figure (new supplementary Figure S1) of the whole dataset of T1 and T2 of non-COVID and COVID groups. It is possible to observe that the heatmap with all individual samples presents the same metabolic clusters and fluctuation profile of the discriminating metabolites of the heatmap with the sample average, indicating that the intensity levels are representative of the population under study.
6-Outliers emerge from the PCA; they suggest poor statistical reliability. I advise authors to review the multivariate approach by applying unsupervised approaches for identifying outliers and supervised approaches for clustering; also validating them through cross-validation.
Response. The removal of outliers is a very important suggestion which we have now addressed in the revised manuscript: new supplementary figure 2A and 2B (PCA scores plot of non-COVID and COVID subjects) and new supplementary table S2, showing the loadings of the metabolites of interest. Please note that the removal of outliers did not change the discriminating pattern between groups, i.e., there was no clear separation between T1 and T2 in both groups, and also the loading factors of the variables of interest were amongst the top 200 values.
We did not proceed to the supervised approach, e.g., PLS-DA, as PCA statistics did not show any clear separation and we wanted to prevent overfitting. Additionally, an important aspect that needs to be clarified is that the variables that were further studied in the univariate analysis were selected according to the PCA loading factors and to the heatmap results, which corroborated the loadings. Additionally, they were all biologically sound and had been previously shown to be involved in the subjects’ responses to SARS-CoV-2 infection. The novelty/strength of our work was to show that there are important changes in these metabolites in the post-acute compared to the acute phase of SARS-CoV-2 infection and the subject’s BMI, viral load and vaccination status contributed significantly to these changes. Importantly, saliva is an important biofluid to address these aspects.
Comments of this nature were included in the revised version of the manuscript (Lines 251-53; 403-04 and 411-12).
7-lines 237-238: Considering the high number of variables, the univariate approach cannot be considered the most appropriate for omics analyses.
Response. We thank the reviewer for addressing this aspect. We agree that univariate statistics as the sole approach could be considered not appropriate for omics studies. However, here we conducted a longitudinal metabolomic study, combining multivariate and univariate analyses, to identify metabolites fluctuation in saliva samples between the acute and post-acute phase of COVID. Using this approach, we intended to describe post-acute alterations in host metabolism that may be associated with long COVID. As mentioned in topic 6, despite the fact that PCA scores plot did not indicate a discriminant pattern between T1 and T2 in both groups, the PCA loading factors, heatmap results, and univariate reinforce the differences found. In agreement, the metabolic signature described in our study has already been described as important in the acute phase and clinical evolution of COVID in others studies. The findings of some studies were described and cited in our discussion. Thus, it is the entire set of analyzes carried out, together with literature, that gives the support of our findings.
8-Line 239: What do the authors mean by "levels" of concentrations?
Response. We thank the reviewer for the careful consideration. For the comparison of each individual metabolite between groups, metabolites’ intensity was used after performing spectral binning. Therefore, the intensity of metabolites is referred to as levels, and not concentrations. For all metabolites, the spectral regions/bins that were chosen for comparison were the ones with the highest intensity and also with no overlapping signals. Additionally, we used CPMG pulse sequence and we scaled all spectra using probabilistic quotient normalization. Therefore, we are confident that we can reliably determine fold changes in metabolites’ intensity using spectral bins.
Comments of this nature were included in the revised manuscript (Lines 150-51, 162-74 and 176-79 of material and methods section).
Why do they not use variables importance projection approach (VIP score)?
Response. We thank the reviewer for addressing this aspect. We did not proceed to PLS-DA - which would provide us with the VIP scores - as PCA statistics did not show any clear separation and we wanted to prevent overfitting. Additionally, as mentioned in the previous comment (topic 6), considering the fact that the heatmap results clearly pointed out the same variables that presented the top 200 loading factors values, we proceed to the univariate statistics using spectral bins to compare T1 and T2 in both groups.
9-Pathway analysis: The Holm p is only significant for the pathway of Glycine. It would be advisable to consider an Enrichment pathway analysis
Response. We thank the reviewer for his/her suggestion. We substitute previous analysis to an Enrichment pathway analysis as suggested but the Holm p is still significant only for the pathway of Glycine. As explained in topic 2, we consider to maintain the analysis in supplementary material and use p-values ​​ < 0.05 only as indicative of pathways in which these significantly altered metabolites between the acute and post-acute phase of COVID would be involved, but including all statistical data. Thus, we specified the parameter used for the statements in the result (line 262).
10-The authors should check the confidence ranges and standard deviations of the graphs in Figure 2 and 3.
Response. We thank the reviewer for raising this concern. We checked the confidence ranges and standard deviations of the indicated graphs. In fact, there is a dispersion of data, which reflects the biological variation of human samples. As mentioned before, the metabolic signature described in our study was supported by statistical analysis and it is in agreement with previous studies of acute and long-COVID, confirming its biological plausibility.
11-Authors are advised to better explain the biological implication deriving from histidine and histamine relationship and to contextualize it with results
Response. We thank the reviewer for his/her suggestion. We included in the discussion section a complete explanation and reference that address the relationship between histidine decrease, observed in our study, with histidine/histamine role in COVID patients (Lines 340-42 and 344-47).
12-Line 372: What do the authors mean by “did not correlate”? A correlation between clinical data and discriminating metabolites would be useful.
Response. We thank the reviewer for addressing this aspect. We agree that the expression “did not correlate” was misused, since no correlation analysis was performed. For viral load we sub grouped the subjects into high and load based on Ct values from qPCR performed in the acute phase, and BCAA level was only increased in subjects with high viral load, while taurine in both groups. For the sake of clarity, we have removed this statement from the text g (lines 389-91).

Reviewer 2 Report
Comments and Suggestions for Authors
The work “Longitudinal 1H NMR-based metabolomics in saliva unveils signatures of transition from acute to post-acute phase of SARS-CoV-2 infection” by Luiza Tomé Mendes and colleagues is a technically sound, well written work investigating the metabolic changes in saliva between the acute and post-acute phases of COVID-19 infection. However, it seems that the authors are overstretching their measures in order to obtain some positive results while there are not there.
The score plots in Supplementary figure 1 do not show any differences between T1 and T2 in COVID and non COVID patients but there are clearly some outlier samples (bottom right on figure S1A and top right on figure S1B). The variables to study further have been selected based on PCA loading plots thus showing the variables responsible for differences in the database, not related to the infection stage. In other words, the outliers may be responsible for BCAA in the loading plots (figure S1C and D respectively). Also, please note that authors state that they use 81498 data points; using a standard p of 0.05 a total of 4000 points would show statistical differences just by chance.
Some suggestions on how to analyse the data
First obtain PCA (as done in the manuscript) and remove possible outliers. Then proceed to to a supervised classification approach (PLS-DA, neural network, …). If the model obtained is statistically different one may proceed to investigate which metabolites are responsible for the differences.
Another option would be to directly evaluate possible differences as done by Luiza Tomé Mendes and colleagues. However, this should be done using metabolite concentrations (please note that Chenomx software used allows to obtain concentrations) but not spectra points or bins.
Author Response
Reviewer 2#
The work “Longitudinal 1H NMR-based metabolomics in saliva unveils signatures of transition from acute to post-acute phase of SARS-CoV-2 infection” by Luiza Tomé Mendes and colleagues is a technically sound, well written work investigating the metabolic changes in saliva between the acute and post-acute phases of COVID-19 infection. However, it seems that the authors are overstretching their measures in order to obtain some positive results while there are not there.
We thank the Reviewer for his/her comments. We would like to clarify the analysis choices performed in order to avoid the interpretation that we are overstretching the results. Here we conducted a longitudinal metabolomic study, combining multivariate and univariate analyses, to identify metabolites fluctuation in saliva simples between the acute and post-acute phase of COVID. Using this approach, we intended to describe post-acute alterations in host metabolism that may be associated with long COVID. The metabolic signature described in our study has already been described as important in the acute phase and clinical evolution of COVID by other studies. The findings of some studies were described and cited in our discussion. Thus, it is the entire set of analyzes carried out, together with literature, that support our findings.
The score plots in Supplementary figure 1 do not show any differences between T1 and T2 in COVID and non COVID patients but there are clearly some outlier samples (bottom right on figure S1A and top right on figure S1B). The variables to study further have been selected based on PCA loading plots thus showing the variables responsible for differences in the database, not related to the infection stage. In other words, the outliers may be responsible for BCAA in the loading plots (figure S1C and D respectively).
Response. Thank you for pointing out our error regarding the description of the PCA results in the previous version of the manuscript. On the other hand, it is important to stress out that in the original submission, the PCA results were included in the supplementary material considering the lack of differences between groups. In the revised manuscript, we stated that PCA statistics indicated that there was not a discriminant pattern between T1 and T2 in both non-COVID and COVID subjects (Lines 235-39).
The removal of outliers was a very important suggestion which we have now addressed in the revised manuscript: new supplementary figure 1A and 1B (PCA scores plot of non-COVID and COVID subjects) and new supplementary table 2, showing the loadings of the metabolites of interest. Please note that the removal of outliers did not change the discriminating pattern between groups, i.e., there was no clear separation between T1 and T2 in both groups, and also the loading factors of the variables of interest were amongst the top 200 values.
An important aspect that needs to be clarified is that the variables that were further studied in the univariate analysis were selected according to the PCA loading factors and to the heatmap results, which corroborated the loadings. Additionally, they were all biologically sound and had been previously shown to be involved in the subjects’ responses to SARS-CoV-2 infection. The novelty/strength of our work was to show that there are important changes in these metabolites in the post-acute compared to the acute phase of SARS-CoV-2 infection and the subject’s BMI, viral load and vaccination status contributed significantly to these changes. Importantly, saliva is an important biofluid to address these aspects.
Comments of this nature were included in the revised version of the manuscript (Lines 251-53; 403-04 and 411-12).
Also, please note that authors state that they use 81498 data points; using a standard p of 0.05 a total of 4000 points would show statistical differences just by chance.
Response. Thank you for noting this. We have now corrected this information as 81,498 data points correspond to the total number of points for all spectra. After processing, some regions and noise threshold were excluded, and then binned with a 0.005 ppm interval. The resulting table presented a final number of 1132 variables/bins. This information was included in the revised version of the manuscript (Lines 169-74).
Some suggestions on how to analyse the data
First obtain PCA (as done in the manuscript) and remove possible outliers. Then proceed to to a supervised classification approach (PLS-DA, neural network, …). If the model obtained is statistically different one may proceed to investigate which metabolites are responsible for the differences.
Another option would be to directly evaluate possible differences as done by Luiza Tomé Mendes and colleagues. However, this should be done using metabolite concentrations (please note that Chenomx software used allows to obtain concentrations) but not spectra points or bins.
Response. We thank the reviewer for the careful consideration. We did not proceed to PLS-DA as PCA statistics did not show any clear separation and we wanted to prevent overfitting. Additionally, as mentioned in the previous comment, considering the fact that the heatmap results clearly pointed out the same variables that presented the top 200 loading factors, we proceeded to the univariate statistics using spectral bins to compare T1 and T2 in both groups. Here, again, some clarification is needed: we used spectral bins - which correspond to metabolites intensity. And in all cases, the spectral regions/bins that were chosen for comparison were the ones with the highest intensity and also with no overlapping signals. Additionally, we used CPMG pulse sequence and we scaled all spectra using probabilistic quotient normalization. Therefore, we are confident that we can reliably determine fold changes in metabolites’ intensity using spectral bins. This information was included in the methodology.

Reviewer 3 Report
Comments and Suggestions for Authors
The authors present a longitudinal study of COVID-19 patients by a non-invasive analysis of saliva. The manuscript is adequately written, but some issues need clarification:
PCA scatter plot does not provide any separation. Considering the large number of buckets, what data reduction method was chosen in Metaboanalyst?
Did the authors exclude any other noise or signal-free regions?
Were the spectra aligned and which method was used to align them?
Author Response
Reviewer 3
The authors present a longitudinal study of COVID-19 patients by a non-invasive analysis of saliva. The manuscript is adequately written, but some issues need clarification:
We thank the Reviewer for his/her analysis of our work. All information linked to your questions were incorporated into the new version of the manuscript.
PCA scatter plot does not provide any separation. Considering the large number of buckets, what data reduction method was chosen in Metaboanalyst?
Response. First we would like to thank the reviewer for pointing out our error regarding the description of the PCA results. On the other hand, it is important to stress out that in the original submission, the PCA results were included in the supplementary material considering the lack of differences between groups. In the revised manuscript, we stated that PCA statistics indicated that there was not a discriminant pattern between T1 and T2 in both non-COVID and COVID subjects (Lines 235-39).
Did the authors exclude any other noise or signal-free regions?
Response. We thank the Reviewer for these comments and we agree that more information is needed regarding spectra pre-processing. Spectra pre-processing was performed in the Metabolab Software (ref. 25), as described in the original submission. Regarding data reduction, in addition to performing bucketing with a 0.005 ppm interval (32 data points), the following regions were excluded:
Signal-free regions: 14.79 to 9.73 and 0.29 to -5.2 ppm
Triton, added for viral inactivation: 0.3 to 0.8, 1.01 to 1.08, 1.52 to 1.71, 2.13 to 2.21, 2.80 to 2.94, 3.54 to 3.72, 6.60 to 6.86, 7.09 to 7.25, 8.26 - 8.61ppm
Ethanol: 1.08 to 1.29 ppm
Water: 5.16 to 4.59 ppm
Additionally, noise filtering was performed and anything below 5x noise threshold-measured between 9.5 and 10.0 ppm region- was discarded.
We included this information in the revised version of the manuscript (Lines 162-75).
Were the spectra aligned and which method was used to align them?
Response. This is a very important parameter and we thank the reviewer for addressing it. Yes, spectra were aligned using the Icoshift algorithm (https://www.sciencedirect.com/science/article/abs/pii/S1090780709003334), which is integrated in the Metabolab software. This information was included in the revised version of the manuscript along with a new reference of Icoshift (Lines 176-79).

Round 2
Reviewer 1 Report
Comments and Suggestions for Authors
The authors have improved the manuscript as suggested.
Author Response
We thank the reviewer for his/her contributions to improving the manuscript.Reviewer 2 Report
Comments and Suggestions for Authors
The revised work by Luiza Tomé Mendes and cols address some of concerns raised by this reviewer but there are some other important aspects of their work that should be addressed before it can be published.
Supplementary PCA figures are the same as in the original version. It is not clear if outlier samples were removed and if the whole spectra of the one without “contaminants” used for the revised version of their work.
It is difficult to understand how it is possible not to find differences in PCA (or discriminant analysis) and then finding them in individual bin by bin analysis. This would suggest that spurious results are a real concern, and authors have to make clear that it is not the case.
I do agree that by using a CPMG pulse sequence it is not possible to obtain absolute concentration values in Chenomx. However, authors MUST measure whole peak area and not small bins of 0.005 ppm as there is a possibility (of 5%) that the results are spurious. This can be done preferably by using a deconvolution software as Chenomx or provided by Bruker or by adding all the bins necessary to account for the area of the whole peak. Changes in metabolite concentration/amount due to various factors analysed should be done on the peak area.
In their letter Mendes and cols state that the variables of interest are within the top 200. What variables/metabolites are the top 10 of the analysis? Do they correspond to metabolites/peaks?
Effective T2 delay –(T2 filter) is missing from the methods section.
Author Response
Response to Reviewer 2#
The revised work by Luiza Tomé Mendes and cols address some of concerns raised by this reviewer but there are some other important aspects of their work that should be addressed before it can be published.
We thank the Reviewer for his/her comments. We would like to apologize to the fact that the revised version of the supplementary material was not uploaded into the system (apparently there was an error in the submission process), which most certainly compromised the second round of the reviewer's comments. Following reviewer # 2 suggestions, in addition to text editing, we included a new supplementary figure (Fig S1), removed outliers from the PCA statistics (Suppl figure 2), and included revised Tables S1 and S2 in supplementary material, containing not only the chemical shift but also the intensity of the peaks (Table S1) and information on the loading factors of the assigned metabolites (Table S2). We therefore ask the reviewer to consider the revised supplementary material into his/her comments.
Supplementary PCA figures are the same as in the original version. It is not clear if outlier samples were removed and if the whole spectra of the one without “contaminants” used for the revised version of their work.
Please check the new version of the supplementary material. We provided a new Supplementary Figure S2, where outliers were removed from the PCA statistics. The removal of outliers did not change the discriminating pattern between groups, i.e., there was no clear separation between T1 and T2 in both groups.
It is difficult to understand how it is possible not to find differences in PCA (or discriminant analysis) and then finding them in individual bin by bin analysis. This would suggest that spurious results are a real concern, and authors have to make clear that it is not the case.
As mentioned in the round 1 of the review, our analyses were not randomly targeted. Despite the fact that PCA did not point to a discriminant pattern between T1 and T2 in both groups, the variables that were further studied were selected according to the PCA loading factors and to the heatmap results and also according to biological soundness. The heatmap clearly pointed out the same variables that presented the highest loading factors values, thus we proceed to the univariate statistics comparing T1 and T2 metabolite levels in both groups. The metabolic signature described in our study was supported by statistical analysis and it is in agreement with previous studies of acute and long-COVID, confirming its biological plausibility.
I do agree that by using a CPMG pulse sequence it is not possible to obtain absolute concentration values in Chenomx. However, authors MUST measure whole peak area and not small bins of 0.005 ppm as there is a possibility (of 5%) that the results are spurious. This can be done preferably by using a deconvolution software as Chenomx or provided by Bruker or by adding all the bins necessary to account for the area of the whole peak. Changes in metabolite concentration/amount due to various factors analysed should be done on the peak area.
We thank the reviewer for addressing this issue and we now have included in the univariate statistics the column assignment (bins) that accounted for the whole peak area, considering non-overlapping peaks. Effectively, the results of the metabolites comparison between groups remain the same.
Confirming our previous results, after recalculation, the same metabolites were statistically different in the non-COVID and COVID group (New Figure 2). In addition, the relationship of fluctuation of metabolites with the overweight, lack of vaccination, high viral load and time to positivity groups were confirmed after reanalysis (New figures 3 and 4).
In their letter Mendes and cols state that the variables of interest are within the top 200. What variables/metabolites are the top 10 of the analysis? Do they correspond to metabolites/peaks?
The assigned metabolites and top 10 of PCA were included in the new version of supplementary material (Table S1 and S2).
Effective T2 delay –(T2 filter) is missing from the methods section.
Considering 32 loop counters and a delay of 0.001s, the effective T2 delay of the transverse relaxation filter was 68.68 ms. This information was included in the revised version of the manuscript (lines 153-54)
